# Development of Novel Oral Delivery Systems Using Additive Manufacturing Technologies to Overcome Biopharmaceutical Challenges for Future Targeted Drug Delivery

**DOI:** 10.3390/pharmaceutics17010029

**Published:** 2024-12-27

**Authors:** Micol Cirilli, Julius Krause, Andrea Gazzaniga, Werner Weitschies, Matteo Cerea, Christoph Rosenbaum

**Affiliations:** 1Department of Pharmaceutical Sciences, University of Milan, GazzaLaB, via Giuseppe Colombo 71, 20133 Milan, Italy; 2Department of Biopharmaceutics and Pharmaceutical Technology, Institute of Pharmacy, University of Greifswald, Felix-Hausdorff-Strasse 3, 17489 Greifswald, Germany

**Keywords:** 3D printing, additive manufacturing, retentive drug delivery systems, personalized drug delivery systems

## Abstract

**Background/Objectives:** The development of targeted drug delivery systems for active pharmaceutical ingredients with narrow absorption windows is crucial for improving their bioavailability. This study proposes a novel 3D-printed expandable drug delivery system designed to precisely administer drugs to the upper small intestine, where absorption is most efficient. The aim was to design, prototype, and evaluate the system’s functionality for organ retention and targeted drug release. **Methods:** The system was created using 3D printing technologies, specifically FDM and SLA, with materials such as PLA and HPMC. The device was composed of matrices and springs, with different spring geometries (diameter, coil number, and cross-sectional shape) being tested for strength and flexibility. A gastro-resistant string was used to maintain the device in a compact configuration until it reached the neutral pH environment of the small intestine, where the string dissolved. The mechanical performance of the springs was evaluated using a texture analyzer, and the ability of the system to expand upon pH change was tested in simulated gastrointestinal conditions. **Results:** The results demonstrated that the system remained in the space-saving configuration for two hours under acidic conditions. Upon a pH change to 6.8, the system expanded as expected, with opening times of 5.5 ± 1.2 min for smaller springs and 2.5 ± 0.3 min for larger springs. The device was able to regain its expanded state, suggesting its potential for controlled drug release in the small intestine. **Conclusions:** This prototype represents a promising approach for targeted drug delivery to the upper small intestine, offering a potential alternative for drugs with narrow absorption windows. While the results are promising, further in vivo studies are necessary to assess the system’s clinical potential and mechanical stability in real gastrointestinal conditions.

## 1. Introduction

The oral route has always been the most common way for drug administration [1,2]. However, there are several challenges associated with this oral drug delivery, such as the limited or highly variable bioavailability of an increasing number of active ingredients [3]. The reasons for this are many and lie in the gastrointestinal tract itself, such as the high variability in the transit behavior of the dosage form caused by the physiology of the GIT in conjunction with the specific absorption windows of the active ingredients, as well as in the solubility and permeability requirements of the drugs themselves [4,5,6]. However, the solubility and permeability are also a challenge [7], as is the stability of the drugs, especially the novel peptide or protein-based drugs [5,8]. Not only are peptide-based drug formulations challenging, but also small molecules, such as metformin, acyclovir, or levodopa, which have a narrow absorption window in the upper small intestine, could benefit from this delivery system. As a result of these challenges, many compounds are currently only parenterally bioavailable, which affects both patient compliance and the regulatory requirements for these dosage forms, as well as their manufacture and cost. A major pharmaceutical and biopharmaceutical challenge is, therefore, to develop novel, robust, and cost-effective oral systems that meet these requirements [9,10,11]. But also, cost-effective development is crucial, and 3D printing offers a significant advantage here by enabling low-cost production with the ability to create innovative and complex designs, which could not be achieved with traditional methods.

The small intestine is the most important organ for the absorption of orally administered drugs due to its villi and microvilli, the resulting very large absorption surface and the particularly pronounced blood circulation [12,13]. The muscular structures enable the transport of food within the small intestine, the pH in the small intestine is neutral and the mucosal surface is permanently moist, with fluid distribution not being homogeneous but in the form of so-called water pockets [14].

In recent years, a variety of capsule- or tablet-based delivery technologies have been developed for targeted drug delivery in the small intestine, including some very creative approaches such as drug-loaded stents, various mucoadhesive delivery concepts or even oral mini-melt extruders [15,16,17]. The mentioned biocompatible stents are able to exert an expanding force not losing their flexibility, being able to deliver drugs, when loaded with some API, and keeping the organ lumen open, while the mucoadhesive systems could be eventually retained in the stomach or in the upper small intestine thanks to their ability to adhere to the mucosa not being affected by organ contractions.

With the impressive development of different additive manufacturing technologies, for example, completely new technological possibilities and in some cases also materials are now available for the development of novel oral drug delivery concepts, allowing a new design based on the white paper method of oral dosage forms completely detached from the conventional capsules and tablets [18,19].

The ability to design and manufacture systems with specific shapes, configurations and complex geometries that interact with the physiology of the gastrointestinal tract in a way that can lead to targeted application in a specific area of the gastrointestinal tract can be made possible using novel technologies such as 3D printing. Hereby, the construction of solid objects of any shape is achieved by adding materials layer-by-layer based on a digital model [20,21,22].

The term “3D printing” includes many techniques that differ from each other in the nature of the substrate, mechanism of layer formation/deposition mode, printer used and, of course, characteristics of the final product [23]. Thanks to the possibility of producing small batches, this technology is also very interesting for personalized and on-demand production, which keeps costs down despite sometimes extremely complex structures [24].

Based on this, the aim of the present work was to design and develop a prototype of a novel drug delivery platform that can control drug release in the upper gastrointestinal tract, especially in the small intestine, and to enable targeted drug release at the small intestinal mucosa. Two different 3D printing techniques were employed to produce the prototype: fused deposition modeling (FDM) and stereolithography (SLA). The novel application technology shown in Figure 1 should contain two matrices loaded with a drug component, such as mucoadhesive films or mini-tablets, connected by a spring-based release mechanism. The spring should press the matrices with the drug components onto the small intestinal mucosa with a defined pressure over a defined length after the previous protective film or thread has dissolved, triggered by a trigger mechanism. The components should preferably be combined with a component that erodes or dissolves under physiological conditions to allow degradation after a defined time and, thus, the safe emptying of the prototype.

The aim of this work was to prepare the conceptually designed application device and to characterize and discuss biorelevant key components such as the spring and trigger mechanism.

## 2. Materials and Methods

### 2.1. Materials

Tough PLA Filament (Ultimaker, Utrecht, The Netherlands), AFFINISOL™ HPMC HME 15LV (Dow Chemicals, Midland, TX, USA), and TPU 95A Filament (Ultimaker, Utrecht, The Netherlands) were used for the 3D printing process by FDM; clear resin V4 (Formlabs, Somerville, MA, USA) was used for 3D printing by SLA. Eudragit L100-55 (Evonik, Essen, Germany), Eudragit L100 (Evonik, Essen, Germany), triethyl citrate (Merck, Darmstadt, Germany), E.S.P. Polyvinyl alcohol (PVA) string Mod. 3, Ply (Drennan International Ltd., Oxford, UK), and EMPROVE^®^ ESSENTIAL PVA 18–88 (PVA 18–88, M_w_ ≈ 96,000 g/mol, Merck KGaA, Darmstadt, Germany) were used for mucoadhesive film, string and glue preparation. Calcium dihydrogen phosphate (JRS Pharma, Rosenberg, Germany), magnesium stearate (Sigma-Aldrich, St. Louis, MO, USA), croscarmellose sodium (JRS Pharma, Rosenberg, Germany), and fumed silica (Caelo, Hilden, Germany) were used for preparation of the placebo tablets. Furthermore, hard gelatin capsules (size AAA, DBcaps^®^ (Lonza, Basel, Switzerland)) were used.

### 2.2. Methods

**HPMC extrusion:** AFFINISOL™ HPMC HME 15LV was extruded by using a bench-top co-rotating, twin-screw extruder (Three-Tec ZE 12, Seon, Switzerland) with four heated zones. A M6 long die having a 2.8 mm diameter was used to extrude the HPMC, the powder was loaded by gravimetric feeding, screw speed was set at 25 rpm. Temperature was set at 80 °C for the zone 1 and 140 °C for zones 2, 3, and 4.

**3D printing of the matrices**: placebo matrices/supports have been designed by using the open-source parametric 3D software Free-Cad (V 0.20.2), and sliced with the Ultimaker Cura program (V 5.3.1, Ultimaker, Utrecht, The Netherlands) for the setting of printing parameters. Objects made of PLA were printed with the Ultimaker 3D fused deposition modeling (FDM) printer (Ultimaker, Utrecht, The Netherlands). Nozzle size used for the printing was 0.4 mm (printhead type AA), printing temperature 215 °C. Infill was 100% and the printing speed 25 mm/s. Objects made of the extruded HPMC K15M were printed with the same FDM 3D printer equipped with a nozzle having diameter 0.4 mm (printhead type AA), setting building plate at 60 °C and extrusion temperature 185 °C; the fan was disabled and no infill geometry was set, but the matrices were made entirely of concentric walls (12 walls). The matrices prepared had dimensions of 22 mm × 7.8 mm × 3 mm. Objects with circular concavities with a diameter of 4 mm, made to host inside small tablets, were also prepared by using the same procedure.

**Tablet preparation:** drug-free tablets were prepared from a powder mixture (calcium dihydrogen phosphate 96%, sodium crosscarmellose 1.0%, magnesium stearate 2.0%, fumed silica 1.0%) using a single-stamp tablet press (KP2, VEB Kombinat NAGEMA) with a flat die, diameter: 3.00 mm.

**Spring design:** springs with different dimensions were designed by using Free-Cad (V 0.20.2). Various geometries were tested to assess the influence of the springs: the coil diameter, the spring diameter, the number of working coils, and the coil geometry (rectangular and circular section) were varied as shown in Figure 2.

A total of thirteen different springs were investigated, which can be categorized into five groups (Table 1). Both circular and rectangular coil geometries were analyzed. The coil diameter varied between 1.0 and 1.5 mm, and the rectangular profile was 1 × 1.5 mm as standard. The springs were analyzed with either four or five coils. The spring diameter varied between 5 and 8 mm.

**FDM 3D printing of the springs.** For preliminary tests, 3D spring models were sliced using the Ultimaker Cura (V 5.3.1) and printed with FDM 3D printer equipped with two nozzle (Mod. 3, Ultimaker, Utrecht, The Netherlands). For 3D printing with PVA supports, the following settings were used: for TPU, printing temperature 225 °C, infill 15%, printing speed 10 mm/s and nozzle AA 0.4 mm were used; for PVA: printing temperature 220 °C, infill 5%, printing speed 10 mm/s and nozzle BB 0.4 were used. For 3D printing without support, the following settings was used: TPU printing temperature 225 °C, infill 25%, printing speed 15 mm/s and nozzle AA 0.4 mm.

**SLA 3D printing of the springs.** The Formlabs PreForm (V 3.30.0) program was used to set printing parameters for the stereolithography printer, as reported here: support density 1.00, contact point dimension 0.20 mm without internal supports. The springs were printed in Clear Resin V4 by FormLabs 3 SLA printer (Formlabs, Somerville, MA, USA) and afterwards washed in isopropanol and cured for 30 min at 375–405 nm (UV curing chamber, XYZPrinting Inc., Lake Forest, CA, USA).

**Mucoadhesive film preparation and gastric resistant string and glue preparation.** All films were produced using the solvent casting technique. Demineralized water (80.0 g) was used as a solvent, anhydrous glycerol (2.0 g) was used as a plasticizer and polyvinyl alcohol (18.0 g) as the base polymer.

The dispersion was then stirred on a stirring plate (IKA^®^ RCT basic, IKA^®^-Werke GmbH and Co. KG, Staufen, Germany) at 150 rpm for 1 h until a clear solution was obtained. If air bubbles were still visible, the polymer mass was centrifuged at 4400 rpm for 15 min (Centrifuge 5702 R, Eppendorf SE, Hamburg, Germany). The bubble-free solution 1 was then applied to a specially coated paper, the so-called release liner, using a doctor blade (mtv messtechnik oHG, Erftstadt, Germany) with a width of 500 µm and a motorized vacuum coating bench at 12.0 mm/s (Automatic Precision Film Applicator CX4, mtv messtechnik oHG, Erftstadt, Germany). The coated laminates were then dried in a fume cupboard at room temperature.

The following formulation (Table 2) was, instead, used to produce the gastric resistant strings (formulation 1) and as the glue to fix system components (formulation 2).

The dispersions were stirred in a glass bottle on a stirring plate (IKA^®^ RCT basic, IKA^®^-Werke GmbH and Co. KG, Staufen, Germany) at 150 rpm for 24 h until a clear solution was obtained.

Commercially available 40 cm long multifilament polyvinyl alcohol fishing line string Mod. 3, Ply (Drennan International Ltd, Oxford, United Kingdom) was coated by a simple dipping process. The acetone used evaporated at room temperature.

**Assembly of the novel application device.** Enteric-retentive systems were prepared by manually assembling and gluing a spring with two PLA units by using the acetonic Eudragit L solution 2 (Table 2). The device was held in the space-saving configuration by wrapping a gastric-resistant string around the system with a compressed spring. Prototypes made with the matrices with the concavities were prepared by pushing two small tablets inside so that they were framed into the matrix. With respect to prototypes having a mucoadhesive film surrounding the system, the foil was put on the system and closed around it by slightly moistening one edge of the film to make it adhesive and capable to be sealed to the other edge.

**Spring strength test.** The elastic strength of the springs was tested with a texture analyzer (LLOYD, TA plus Ametek, United Kingdom), and the associated software, Nextgen Plus, version 3.0 (Ametek, Berwyn, PA, USA), with the following settings: displacement speed at 10 mm/min, load cell of 10 N (upper limit). The test was run by selecting a maximum load of 1.4 N.

**Evaluation of opening time of enteric-retentive systems.** Systems, maintained in their space-saving configuration by the gastro-resistant string wrapped around the device, were placed in 800 mL of USP 44 hydrochloric acid buffer pH 1.2 at 37 °C and under magnetic stirring (130 rpm) for 120 min. After this time, fluid was replaced with 800 mL pH 6.8 phosphate buffer (PBS pH 6.8, according to USP). To protect the system from the magnetic stirrer (Figure 3), a distance mesh insert designed and 3D-printed, as described by Großmann, L and colleagues, was used [23]. The evaluation of the opening time of the systems was performed in triplicate for each spring configuration.

## 3. Results and Discussion

Different geometries of the expanding device were designed according to the purpose of the system and the desired drug release time. As a result, different drug containing units such as films, minitablets and hydrophilic matrices were considered. In addition, springs of different lengths, shapes, and strengths were manufactured to meet specific requirements. The matrices were designed in a classic oblong shape with a convex top and a flat bottom with a small concavity for the spring. Alternatively, 22.0 mm × 7.8 mm × 3.0 mm PLA matrices (carriers) were printed in the same shape but with two concavities on the top to hold two drug-containing minitablets (Figure 4). Depending on the purpose of the device, various modifications and characteristics of its components can be adjusted, such as springs of different sizes and strengths. The system must reach the organ, expand and withstand contractions. Accordingly, different drug-containing units or delivery devices can be tailored depending on the absorption site and drug dose. Various prototypes of devices equipped with drug-containing matrices or units designed for the delivery of small tablets or films have been developed. These tablets and films are made of polymeric materials that form gels with mucoadhesive properties that enhance adhesion to the internal surface of the organ.

### 3.1. FDM Printing Realization of the Matrices

Matrices were produced in different configurations and materials to have the possibility to vary depending on the purpose of the system. Objects made of PLA and having a concavity in the flat part to permit the positioning of a spring were designed and printed and their possible utilization would be to be supports for a drug loaded film that can be applied to the external surface of such objects. Similar objects were printed in HPMC so that hydrophilic matrices were obtained with the idea of producing drug containing mucoadhesive matrices. The used HPMC is a common excipient for tablets with various release kinetics, depending on whether it is used as a coating or within a matrix. It can also be processed into an oral dissolving film; in which case it may exhibit mucoadhesive properties. Furthermore, HPMC is an ideal candidate for a first proof of concept, as it can be effectively processed in both conventional melt extrusion systems and 3D printing. In contrast, PLA is not soluble but is also described as a pharmaceutical excipient. In this case, it was used in technical grade, as it is the simplest, most standard material to process and the most cost-effective option for 3D printing. Finally, a further design was printed having two housings on the upper surface especially made to be able to contain two small mucoadhesive tablets. Work was focused on the design and printability of such shapes, therefore no drug was loaded into such prototypes.

### 3.2. Preparation of Springs Using 3D Printing and Mechanical Evaluation

Springs having a conical shape with a small space-saving geometry when compressed were designed. However, this shape did not allow the production of springs having the desired high and diameter (Figure 5C,D). Springs with cylindrical geometry were therefore designed in different shapes and dimensions of the section, number of coils and diameter (Table 1) to evaluate their relationship between those parameters and their strength and flexibility. Preliminary trials were performed by printing springs by means of an FDM 3D printer and by using filament of TPU 95A as an elastic polymer (data not presented) (Figure 5C,D) resulted in prototypes with poor definition and low mechanical resistance. To obtain objects with a good reproducibility and a good resolution, an SLA 3D printer was selected using V4 Clear resin as printing material (Figure 5A,B). Results with this technique have been satisfying in terms of precision and resolution, and suitable for the printing of springs and units. Springs having different characteristics in terms of cross-section geometry, cross-section dimensions, number of coils and their diameter, have been designed and produced for evaluating their mechanical properties.

The choice of materials was primarily guided by the 3D printing process and the specific shaping capabilities each technique provides. For FDM printing, a standard PLA filament was used, with polyvinyl alcohol serving as the support material. For the SLA printer, the standard resin supplied with the printer was utilized for initial evaluation. Looking ahead, the field of resin-based printing offers a vast array of polymers, compositions, and material properties that could be explored in future studies. These alternatives hold significant potential to enhance the device’s performance or enable the creation of more compact designs. This is particularly relevant given the high resolution of SLA printing and the nearly limitless geometric possibilities it affords.

To evaluate the mechanical properties, springs with clamps were designed (Figure 5A). As there are no data on how much force a spring can apply to the support without causing pain to the patient, the spring was tested in the Texture Analyzer. The maximum force was set at 1.4 N based on Smart Pill data from studies with healthy volunteers [25]. The resulting graphs are reported in Figure 6 and Figure 7.

The recorded force-displacement diagrams show slight deviations in the resulting force for compression and return journeys. As expected, springs with a smaller diameter, fewer coils, and a bigger cross-section dimension appear to be harder, while those with a bigger diameter, more coils, and a smaller cross-section diameter result weaker/softer. Springs with a rectangular section of 1 × 1.5 mm showed a behavior like the spring having the same characteristics but a circular section cross-section of 1.5 mm, meaning that section shape (circular or rectangular) leads to no difference in terms of the spring’s strength. Spring C.CS1.C4.C5, having the smaller cross-section, was not able to generate the maximum value of 1.4 N event when reached the maximum compression displacement at 13.5 mm. In particular, the spring with a diameter of 8 mm showed a strong increase in force after 10 mm of compression, which can be attributed to the contact between coils and thus to the maximum deflection of the spring (Figure 6C). The rectangular profile springs behaved in the same way as the round springs (Figure 7). Elasticity modulus for resin V4 employed in the 3D printing of the spring was reported to be 2.8 GPa (2.8 × 10^9^ N/m^2^) [26]. Mechanical properties of the springs are then given also by diameter, cross-section geometry, and diameter, length of the spring and number of coils, as described by Hooke’s law equation.

### 3.3. Assembly and Evaluation Novel Dosage Form

For enteric-retentive systems an additional holding string having gastric-resistant properties was wrapped around the device in the space-saving configuration before placing the systems inside the body of a hard gelatine capsule (Figure 8). The gastric-resistant filament maintains the device in the space-saving configuration. The dissolution of the holding filament occurs in the intestinal lumen where pH turns relatively less acid, and the device is expected to expand immediately and gain a dimension that is compatible with the internal lumen diameter of small intestine, eventually allowing the drug containing units to adhere to intestinal walls.

Based on the results obtained with the measurements of the spring strength, systems consisting of a spring and two drug-delivering units were assembled by gluing the components (spring and two units/supports) using an adhesive acetonic solution prepared dissolving a polymer having pH-dependent solubility (Eudragit L100), soluble at pH above 5. Systems based on springs with five coils with a circular section of 1 mm or 1.5 mm diameter and a coil diameter of 5 mm or 7 mm (C.CS1.C4.D5 or C.CS1.5.C4.D7, respectively) were chosen for testing. The springs were selected based on a minimum strength observed on the string with a strength comparable to the values of the pressure exerted in the intestinal tract.

Since the enteric-retentive system is not meant to remain assembled in the intestine for a long period, but only for approximately 15 min, it is important that the device is able to withstand the dominant contraction frequency of approximately 12 cycles per minute and a slow-wave propagation velocity of 15 cm/min [27]. However, the systems also need to be removed promptly from the space-saving configuration compatible with the oral administration to the expanded form, able to maintain the device in place and allow the release of the drug. For enteric-retentive systems, the space-saving configuration was obtained by wrapping the device with a gastric resistant string prepared by coating a water soluble PVA string with a gastric-resistant polymer, Eudragit L100-55, a polymer with pH-dependent solubility, soluble at pH above 5.5.

To assess of the opening time of the enteric-retentive devices, systems forced in their space-saving configuration by the gastro-resistant string were placed pH 1.2 buffer for 120 min; after this time, aqueous media was changed for pH 6.8 phosphate buffer to mimic gastrointestinal transit.

Results showed that the gastro-resistant strings were able to resist with no detectable modifications for two hours in acidic conditions and to maintain the system in a space saving configuration. Opening times after pH change resulted in 5.5 ± 1.2 min for systems prepared with springs with a diameter of 5 mm and a section of 1 mm and 2.5 ± 0.3 min for those prepared with springs of a 7 mm diameter and 1.5 mm section. As expected, the system prepared with springs with a smaller diameter section opened significantly more slowly than those with thicker springs. The pressure was clearly related to the strength of the spring previously evaluated with the texture analyzer. In both cases, systems were disassembled in less than 60 min ± 10 min, thanks to the dissolution of the glue made of Eudragit L100 used to attach the springs to the supports. This disassembling process could be essential to ensure the safe evacuation of the device from the gastrointestinal tract. The dimensions of the system before being forced in the space saving configuration were recovered after opening, as demonstrated by measures reported in Table 3. Springs, and consequently the whole systems, were able to return to their original shape after being maintained in aqueous fluids and compressed in the space saving configuration for two hours.

However, it should be noted that the device was not subjected to mechanical stress in our in vitro test, as is often the case with oral dosage forms in the gastrointestinal tract. The mechanical stability of the device, which may swell during gastric passage, needs to be investigated in further studies. It is also of great interest to simulate further studies on the residence time of the device, e.g., after postprandial ingestion, in a biorelevant way, to assess the behavior of the dosage form, which in such a case is likely to have already been released in the stomach and may disintegrate into its components due to the postprandial environment. The device represents a potential trigger mechanism that remains stable in the stomach and only activates in the small intestine. However, it is important to emphasize that this is an early prototype design that must undergo further development and refinement in subsequent stages. This device could only be made possible through rapid prototyping, which provided a straightforward and cost-effective method, offering significant advantages during the early stages of development. The ability to quickly iterate on designs is crucial for exploring novel concepts.

While this approach has proven invaluable at the prototype stage, it is likely that other manufacturing techniques, such as injection molding, may be considered in the future as the device progresses. These decisions will be addressed during design reviews, depending on the scalability and performance requirements. The interest in making previously parenterally administered drugs orally bioavailable is immense. Unlike regular injections, this could significantly improve patient compliance and reduce the complications associated with frequent injections. Moreover, the potential of this device to target specific areas within the gastrointestinal tract opens new possibilities for highly localized drug delivery, ensuring more efficient treatment with fewer side effects. Looking ahead, further research into the precision of targeting and controlled release will be crucial for enhancing therapeutic outcomes. However, it is essential to thoroughly evaluate the safety, functionality, and concept validation of the device through in vivo studies before moving forward.

## 4. Conclusions

A new protype concept for a 3D-printed expandable drug delivery system was proposed to administer active ingredients having narrow absorption window exactly to the anatomical site in which they are absorbed. This system is still in the prototype phase and requires further validation. 3D-printed systems intended for organ retention have been designed, size expansion strategy has been employed and springs have been selected as expandable units. Different springs in terms of diameter, number of coils and cross-section have been produced and tested to evaluate their behavior when a strength comparable to the organ contractions was exerted on them by a texture analyzer. As expected, different configurations of the spring led to different strengths and, therefore, hypothetical different purposes of the systems depending on the exhibited resistance. The results showed that the system can remain in space-saving configuration for two hours into acidic conditions, thanks to a gastro-resistant string enveloping the system, and then expanded in pH 6.8 when the string dissolved due to pH change. Since the work was focused on small intestine delivery of the drug, the results obtained in terms of ability of resisting the acidic environment and regaining the expanded configuration afterwords are promising and make this prototype interesting for upper intestine targeted drug delivery. These findings make this prototype an interesting candidate for upper intestine-targeted drug delivery, though further validation and refinement are needed to fully assess its clinical potential.

## Figures and Tables

**Figure 1 pharmaceutics-17-00029-f001:**
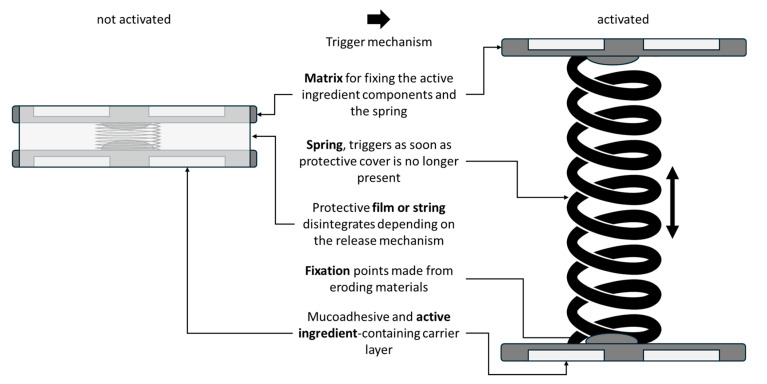
Concept sketch of the spring-like delivery device and initial CAD interpretation of the same.

**Figure 2 pharmaceutics-17-00029-f002:**
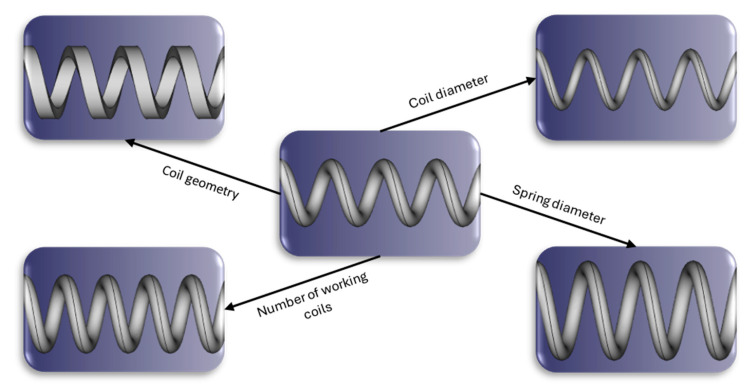
Schematic illustration of the different design alternatives considered for modifying the mechanical properties of the springs used.

**Figure 3 pharmaceutics-17-00029-f003:**
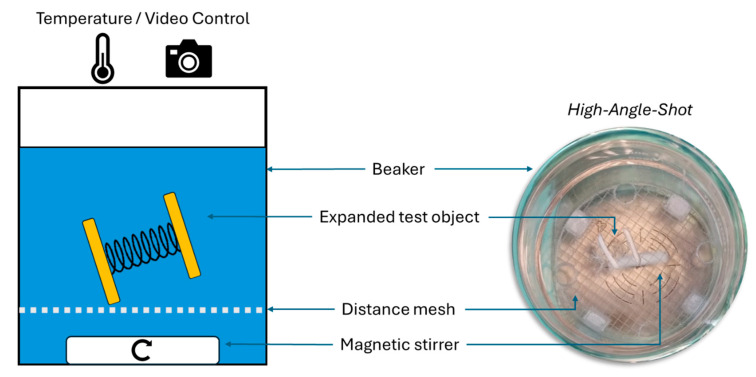
Sketch of the test set-up to investigate the trigger mechanism of the novel, additively manufactured, spring-based application system.

**Figure 4 pharmaceutics-17-00029-f004:**
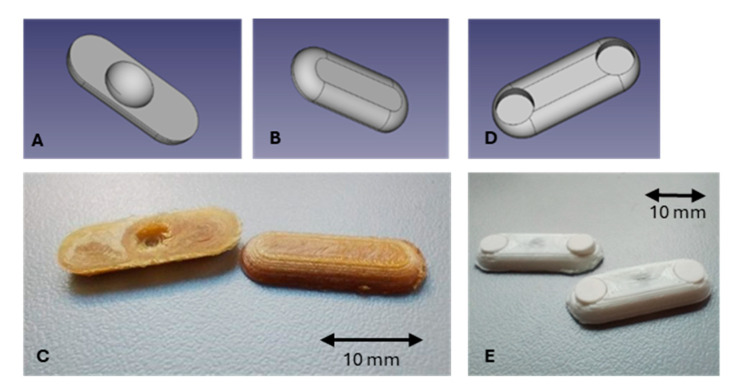
Designs (**A**,**B**,**D**) and images of prototypes (**C**,**D**) prepared using 3D printing, including matrices (**C**) and supports for minitablets (**E**).

**Figure 5 pharmaceutics-17-00029-f005:**
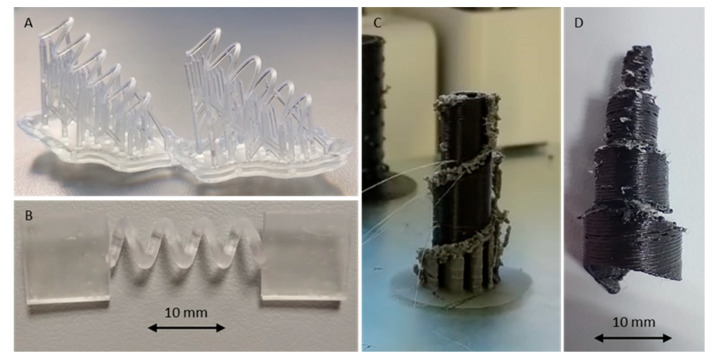
(**A**): SLA 3D-printed spring. (**B**): SLA-printed spring with support for mechanical testing. (**C**): FDM-printed spring with a two-component printing system using water soluble PVA as a support. (**D**): Cleaned FDM-printed TPU spring.

**Figure 6 pharmaceutics-17-00029-f006:**
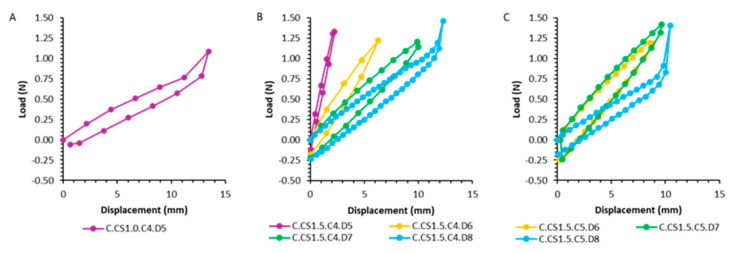
Force-displacement diagrams for cylindrical springs: (**A**): Small coil diameter of 1 mm. (**B**): Variation of the spring diameter from 5 to 8 mm. (**C**): Increasing the number of working coils and varying the coil diameter.

**Figure 7 pharmaceutics-17-00029-f007:**
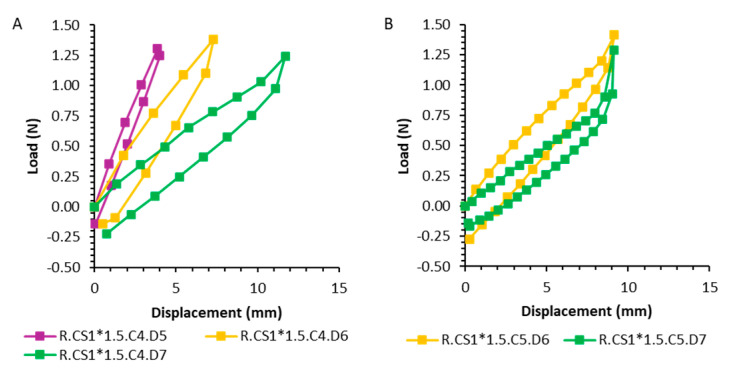
Force-displacement diagrams for cylindrical springs with rectangular cross-section: (**A**): Variation of the spring diameter. (**B**): Variation of the spring diameter with an increased number of working coils.

**Figure 8 pharmaceutics-17-00029-f008:**
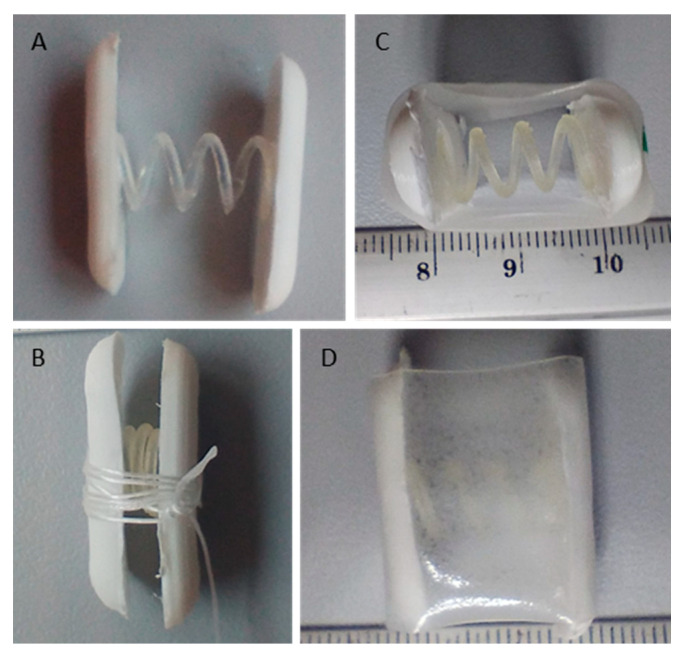
Overview of the different construction states and compositions of the application device. (**A**): PLA matrix with spring with thin coil diameter. (**B**): PLA matrix with spring with thin coil diameter with a gastric juice-resistant thread as trigger mechanism in central formation. (**C**): PLA matrix with enteric film as protection so that the trigger mechanism can trigger in a pH-controlled manner. (**D**): Side view of (**C**).

**Table 1 pharmaceutics-17-00029-t001:** Design properties and dimensions of springs prepared.

Code	Coil Section Geometry	Coil Diameter (mm)	Coils Number	Spring Diameter (mm)
C.CS1.0.C4.D5	Circular	d = 1.0	4	5
C.CS1.5.C4.D5	Circular	d = 1.5	4	5
C.CS1.5.C4.D6				6
C.CS1.5.C4.D7				7
C.CS1.5.C4.D8				8
C.CS1.5.C5.D6	Circular	d = 1.5	5	6
C.CS1.5.C5.D7				7
C.CS1.5.C5.D8				8
R.CS1*1.5.C4.D5	Rectangular	1 x 1.5	4	5
R.CS1*1.5.C4.D6				6
R.CS1*1.5.C4.D7				7
R.CS1*1.5.C5.D6	Rectangular	1 × 1.5	5	6
R.CS1*1.5.C5.D7				7

**Table 2 pharmaceutics-17-00029-t002:** Polymer blends to produce gastric resistant string preparation and gastric resistance glue.

Formulation	Polymer		Plasticizer	Solvent
	Eudragit L100-55	Eudragit L100	Triethyl citrate	Acetone
1	20.0 g		4.0 g	76.0 g
2		20.0 g		

**Table 3 pharmaceutics-17-00029-t003:** Measures of enteric-retentive systems in the expanded and space-saving configuration before and after opening in pH 1.2 buffer for 120 min, and pH 6.8 buffer for the remaining duration of the test.

	Expanded Before Test	Closed Before Test	Expanded After Test	System’s Opening Time at pH 6.8
Diameter 7 mm, Section 1.5 mm	24.1 ± 0.1 mm	11.2 ± 0.3 mm	24.1 ± 0.2 mm	2.5 ± 0.3 min
Diameter 5 mm, Section 1 mm	22.0 ± 0.2 mm	9.1 ± 0.1 mm	22 ± 0.2 mm	5.5 ± 1.2 min

## Data Availability

Data will be made available on request.

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
