# Peer review of "Development of Novel Oral Delivery Systems Using Additive Manufacturing Technologies to Overcome Biopharmaceutical Challenges for Future Targeted Drug Delivery"

_pharmaceutics, 2024, doi:10.3390/pharmaceutics17010029_

Round 1
Reviewer 1 Report
Comments and Suggestions for Authors
The authors need to address the following comments;
General comments:
1. Include recent literature in the introduction part
2. The whole manuscript should be checked for typographical errors
3. Studies such as biodegradation of devices can be included
4. The author can discuss more about the palatability and swellability of the designed device
5. What is the stability of the device in storage condition
6. Please discuss on shape and size regulatory guidelines in terms of the size and shape of the developed device
Specific comments:
1. Improve the resolution of Figure 1
2. Table 2. Please correct typos
3. Line 249, correct the sentence
4. Line 253: correct the title
5. The author can elaborate on future perspective in terms of targeted drug delivery
Comments on the Quality of English LanguageAuthors need to correct language throughout the manuscript
Author Response
Dear Reviewer,
We would like to express our sincere gratitude for your comprehensive and constructive comments on our manuscript. Your feedback and suggestions have been invaluable in enhancing the quality of our work.
We have carefully revised the manuscript and are confident that it now aligns better with your expectations and the high standards of the journal. With this work, we aim to highlight the vast potential of 3D printing in designing innovative drug delivery devices with biopharmaceutical relevance. This approach allows for the development of advanced geometries that could provide significant biopharmaceutical advantages in drug administration, particularly through targeted release in biorelevant regions of the gastrointestinal tract—a feat that is difficult to achieve with conventional tablets or capsules.
This approach represents an "out-of-the-box" perspective on formulation development. While we acknowledge that comprehensive clinical studies are still required to validate these concepts, the unique possibilities offered by 3D printing for dosage form design hold great promise.
We are excited to continue exploring this creative and innovative field and look forward to advancing it further in the future.
Our detailed responses to your comments, along with a description of the changes made, can be found in the attached PDF document.
We hope that the revised version of our manuscript meets your approval and remain available for any further questions or suggestions.
Thank you again for your time and support!

Reviewer 2 Report
Comments and Suggestions for Authors
The manuscript describes an interesting system for the oral delivery of complex active compounds. The delivery device was created with additive manufacturing technology. Although the topic is very relevant, I have concerns regarding the scope of the manuscript. According to the instructions for authors, the Pharmaceutics journal accept two types of manuscripts. A manuscript classified as “article” should report scientifically sound experiments and provide a substantial amount of new information. The manuscript should be revised and improved to include systematically designed experiments including drug delivery tests to prove the new concept of the novel oral delivery device. The experimental protocol should be clearly described in the revised manuscript.
Author Response

(The authors gave the same response as above.)

Reviewer 3 Report
Comments and Suggestions for Authors
The manuscript supplied for review explores the potential of additive manufacturing (3D printing) to create advanced oral drug delivery systems, focusing on targeted release in the upper gastrointestinal tract. The research addresses biopharmaceutical challenges, particularly with drugs in BCS classes III and IV, which exhibit poor permeability and stability, often requiring parenteral administration. The research successfully demonstrated a 3D-printed drug delivery prototype capable of overcoming challenges in targeted oral administration. The system’s ability to expand and adhere to specific intestinal sites offers significant promise for delivering drugs with narrow absorption windows.
The authors should be commended for such a novel approach and design but there is a serious gap in the research presented which would need to be addressed prior to acceptance for publication.
Material Selection - The author should attempt to provide a justification as to why specific materials like PLA, HPMC, and resin were selected, focusing on their biocompatibility, mechanical properties, and ease of use in additive manufacturing. This can be included in the introduction if necessary with supporting references.
Cost effectiveness - While the manuscript mentions cost-effectiveness (as an opening statement), there is a lack of detailed analysis of the cost implications of producing these devices on a large scale compared to traditional formulations.
Stability - The long-term stability of the device during storage and its robustness during gastric transit were only briefly touched on. Real-world usage scenarios should be discussed. Similarly stability was highlighted in the abstract and introduction as key benefits of this approach but not explored during analysis.
Regulations and In Vivo Validation - The manuscript relies on in vitro testing without a roadmap for validating the system in vivo or addressing potential translational challenges or how the proposed device would align with regulatory requirements for novel drug delivery systems or the scalability of manufacturing processes. At minimum a discussion of these barriers and strategies to overcome them would be required in any resubmitted document. It is perfectly acceptable to acknowledge gaps in the study, such as the absence of drug-loading experiments or detailed pharmacokinetic data but any reader of the manuscript should be made aware of these gaps.
Drug Release Kinetics - While the manuscript mentions the system’s opening time, it lacks data or discussion on how the drug release profile would be controlled or optimised. What type of formulation would the drug substance require?
Other topics that should be covered include the potential impact of the device on patient compliance compared to injectable or conventional oral therapies and how advances in additive manufacturing, such as multi-material printing or bio-printing, could further enhance the functionality of the device.
Comments on the Quality of English LanguageThe quality of English in the manuscript is generally good but could benefit from moderate revision to enhance clarity, precision, and consistency. The language is functional and conveys the core ideas effectively; however, there are areas where improvements in grammar and phrasing would help to improve the manuscript.
As well as requiring a general review and proofreading I have highlighted some specific areas that can be addressed directly.
Line 86 - should conceptually described application device be "conceptually designed application device"
Line 351 - 60 min +/- 10 should possibly read 60 minutes +/- 10 minutes
Line 381 - should read small intestine delivery of the drug
Author Response

(The authors gave the same response as above.)

Round 2
Reviewer 1 Report
Comments and Suggestions for Authors
Authors have addressed all the comments, now this work can be accepted in the present form
Reviewer 2 Report
Comments and Suggestions for Authors
The authors have addressed all my concerns, and the manuscript has been substantially revised. In my opinion, the manuscript should be accepted for publication.